# Insect Feeding on *Sorghum bicolor* Pollen and Hymenoptera Attraction to Aphid-Produced Honeydew

**DOI:** 10.3390/insects13121152

**Published:** 2022-12-14

**Authors:** Karen R. Harris-Shultz, John Scott Armstrong, Michael Caballero, William Wyatt Hoback, Joseph E. Knoll

**Affiliations:** 1USDA-ARS, Crop Genetics and Breeding Research Unit, 115 Coastal Way, Tifton, GA 31793, USA; 2USDA-ARS, Wheat, Peanut and Other Field Crops Research Unit, 1301 N. Western Rd, Stillwater, OK 74075, USA; 3Department of Entomology and Plant Pathology, Oklahoma State University, 127 NRC, Stillwater, OK 74078, USA

**Keywords:** agroecosystem, carbohydrate, beneficial insects, conservation biological control, pollen, pollinivore

## Abstract

**Simple Summary:**

Insect pollinators and beneficial insects are in decline worldwide as agricultural practices shift to monocultures of the same crop grown over large areas. In response to the decline, the growth of nectar-rich plants is promoted, while the potential of grasses to provide resources for pollinators is rarely investigated. Sorghum, a widely grown crop produces abundant pollen. Additionally, infestations by the sorghum aphid, *Melanaphis sorghi*, produce large amounts of honeydew, a sugary substance that could provide an alternative to nectar. This field study characterized the use of sorghum pollen and sorghum aphid honeydew by Hymenoptera (bees, parasitoids, wasps) and other insects. Our studies show that susceptible sorghum could provide a food source to at least 29 families of Hymenoptera and other beneficial insects and could be promoted as a valuable landscape planting for preserving these insects.

**Abstract:**

Pollinators are declining globally, potentially reducing both human food supply and plant diversity. To support pollinator populations, planting of nectar-rich plants with different flowering seasons is encouraged while promoting wind-pollinated plants, including grasses, is rarely recommended. However, many bees and other pollinators collect pollen from grasses which is used as a protein source. In addition to pollen, Hymenoptera may also collect honeydew from plants infested with aphids. In this study, insects consuming or collecting pollen from sweet sorghum, *Sorghum bicolor*, were recorded while pan traps and yellow sticky card surveys were placed in grain sorghum fields and in areas with Johnsongrass, *Sorghum halepense* to assess the Hymenoptera response to honeydew excreted by the sorghum aphid (SA), *Melanaphis sorghi*. Five genera of insects, including bees, hoverflies, and earwigs, were observed feeding on pollen in sweet sorghum, with differences observed by date, but not plant height or panicle length. Nearly 2000 Hymenoptera belonging to 29 families were collected from grain sorghum with 84% associated with aphid infestations. About 4 times as many Hymenoptera were collected in SA infested sorghum with significantly more ants, halictid bees, scelionid, sphecid, encyrtid, mymarid, diapriid and braconid wasps were found in infested sorghum plots. In Johnsongrass plots, 20 times more Hymenoptera were collected from infested plots. Together, the data suggest that sorghum is serving as a pollen food source for hoverflies, earwigs, and bees and sorghum susceptible to SA could provide energy from honeydew. Future research should examine whether planting strips of susceptible sorghum at crop field edges would benefit Hymenoptera and pollinators.

## 1. Introduction

Pollination is the act of transferring pollen between male and female parts of flowers, allowing for fertilization and reproduction [1]. Pollination can occur by wind, water, birds, mammals, and insects. One third of the world’s crops, including animal products, is derived primarily from bee pollination [2,3]. In most large-scale agricultural systems, crops are planted as monocultures with one cultivar planted over a large area and as a result have nearly simultaneous development and flowering. As a result of these practices, native and introduced pollinators often suffer from a lack of floral resources, including nectar and pollen, throughout the year. Extensive research has documented population reductions of native fauna when floral sources are scarce. The majority of findings show that a diversified plant community supports more pollinators and far more natural enemies [4,5,6].

Recommendations for supporting pollinator populations promote the growth of nectar-rich plants, yet rarely promote wind-pollinated plants [7] and statements that grasses provide no values to bees or other pollinators appear regularly in the literature [7,8]. However, pollinators collect pollen from many grasses, including row crops and turfgrasses. In particular, bees and syrphid flies, have been documented collecting pollen from at least 99 grass species [7,9]. Additionally, many insects including *Apis mellifera* (Hymenoptera: Apidae, honeybee), *Spatunomia* spp., *Patellapis* (Zonalictus) sp., *Lipotriches* sp., *Nomia* (Acunomia) sp., and *Lasioglossum* (Evylaeus) sp. have been documented collecting pollen from sorghum, *Sorghum bicolor* [10,11,12] Sorghum is a grass planted on 2.95 million hectares (7.3 million acres) in 2021 in the U.S. [13] and is used for food, feed, forage, bioenergy, and syrup production.

In addition to collecting pollen from sorghum, honeydew, the waste-product of Hemiptera feeding on grasses may also provide important sugar resources to pollinators and natural enemies. Honeydew is often considered as plant damage because it creates a beneficial medium for sooty mold to grow, further hindering the plant’s photosynthetic potential [14]. However, honeydew contains many different proteins, sugars, and free amino acids and has been documented to be used by many Hymenoptera, including bees and wasps [15,16]. Additionally, parasitoid species in agricultural systems are also known to use honeydew resources [17] and under laboratory conditions have greater survival times when honeydew is utilized. In addition to providing a food resource, honeydew can act as a kairomone, allowing parasitoids to locate hosts [17].

The sorghum aphid (SA), *Melanaphis sorghi*, (previously known as sugarcane aphid, *Melanaphis sacchari*) has been an economically damaging pest on sorghum since it was first found in 2013 on grain sorghum near Beaumont, TX [18]. By 2015 it had spread to almost all sorghum growing areas in the U.S. [14]. Hymenoptera and Diptera are often observed in sorghum fields infested by SA with many types visiting areas of accumulated honeydew, especially on the tops of leaves beneath infestations of SA [19] (Figure 1). On susceptible sorghums, the SA displays high feeding activity and reproduction—and under favorable conditions—high infestations consisting of tens of thousands of aphids per leaf result in abundant honeydew accumulation [14].

We conducted two studies to (1) identify insects that collect or consume sorghum pollen and determine if the plant height, panicle length and rating date influenced insect abundance and (2) compare the diversity and abundance of Hymenoptera in sorghum fields and Johnsongrass stands infested by SA. We identified the insects collecting or consuming pollen in a sweet sorghum population in Tifton, GA, USA. We conducted a field study to test the effect of SA on Hymenoptera occurrence in sorghum and Johnsongrass plots near Perkins, OK, USA and in Stillwater, OK, USA. We found that several beneficial insect species collect sorghum pollen and that sorghum heavily infested by SA greatly increases the abundance and diversity of Hymenoptera.

## 2. Methods and Materials

### 2.1. Study 1: Sweet Sorghum and Pollen Feeding

#### 2.1.1. Plant Material

Fifty (N109A × PI 257599) F_4_ lines and the parents N109B and PI 257599 were planted in a randomized, complete block design with three replications at the USDA-Belflower Farm (31°30′26.3″ N, 83°33′21.6″ W) in Tifton, GA on 15 June 2021. The parent N109B is a sweet, early maturing dwarf line [20] and PI 257599 is a sweet, late maturing tall line [21,22] and the F_4_ lines were segregating for a large number of traits including plant height, panicle length, and flowering time. The test was planted on a 0.79 ha field, flanked by 149 plots of a (N109A × PI 257599) F_4_ population for advancement and only these 149 plots were sprayed with the insecticide Sivanto^®^ (17.09% flupyradifurone; Bayer Crop Science, Whippany, NJ, USA). Twenty-four plots of breeding lines were also planted to be phenotyped for SA resistance. These tests were surrounded by a border of 772 plots of dwarf, early maturing commercial grain sorghum hybrid NK8416 (S&W Seed Company; Longmont, CO, USA) and 37 plots of Tx2783, an SA-resistant grain sorghum. The border provided additional sources of pollen after sorghum flowering because the border plants produced tillers that flowered until 28 September. Each plot was 4.3 m long, separated on the ends by 1.8 m alleys. Rows were 0.9 m apart. Unusually wet weather (Appendix A) caused an epizootic and SA populations did not establish, but anthracnose disease, caused by *Colletotrichum sublineola*, was prevalent [23].

#### 2.1.2. Insect Identification and Plant Morphological Measurements

Each week, flowering notes were taken on each plot. Once all three replicates of a line were at anthesis, insect data were recorded the next day. Three flowering panicles were selected at random in each plot. For each focal plant, only the primary panicle was selected, excluding the panicles produced from the tillers. The number of insects collecting pollen or consuming pollen was recorded between 8:00 and 10:30 for one minute per panicle. Data were recorded weekly over the course of four weeks from 14 September 2021 to 4 October 2021. Insect data were not recorded if it was raining, and insects moving between panicles in a plot were only recorded once.

At maturity, plant height was measured from the soil to the top of the panicle for three plants per plot. Panicle length was recorded from the base of the panicle to the top for three plants randomly selected per plot. N109B, the other parent of the mapping population, was measured for panicle length and plant height; however, no insect data were measured as it flowered (19 August–1 September 2021) before our insect observations were made.

#### 2.1.3. Analysis

All pollinator visitation data were analyzed in the GLIMMIX procedure of SAS v. 9.4 (SAS Institute, Cary, NC, USA). To test for differences among lines, plant morphological data (plant height and panicle length), and log-transformed insect counts [*y* = log(*x* + 1)] were analyzed by analysis of variance (ANOVA) with line as the only fixed effect. Replication and flowering date were included as random effects. Differences among lines were calculated using Tukey’s HSD at α = 0.05. Among insects—only hoverflies, total bees (honeybees + bumblebees), and total insects had enough non-zero observations for analysis. Because insect counts mostly consisted of ones and zeros, data were also analyzed using a binary logistic model to compare the log-odds of observing an insect on a sorghum panicle on different dates. The data were coded as ‘1’ being the observation of one or more insects and ‘0’ being the observation of no insects. Because each individual line only flowered on a specific date (i.e., date and sorghum line are confounded), date was the only fixed effect in the model. Replication and line (nested within date) were included as random effects. The log-odds were converted to probabilities using the ILINK option in the LSMEANS statement, and differences were calculated using Tukey’s HSD at α = 0.05.

### 2.2. Study 2: Aphid Honeydew and Hymenoptera Diversity

#### 2.2.1. Sorghum

To experimentally test the effect of SA infestation and honeydew production on Hymenoptera occurrence, sorghum plots were established at the Cimarron Valley Research Station located near Perkins, OK, USA (35°59′10.919″ N, 97°3′0.277″ W). Two fields, approximately 0.48 km apart, were chosen, one with SA and the other as a control.

In the experimental field, four rows (approximately 50 m long) of susceptible cultivars (Tx7000) were planted and each row was flanked by a resistant cultivar (Tx2783) with 3.05 m of space between each experimental row. Sorghum was not sprayed with insecticide. Four locations, spaced 10 m apart, were selected in each row. At each location, three different colors of pan traps (blue, yellow, and white) consisting of 350-mL (12 oz) plastic bowls were used to sample Hymenoptera. The traps were half-filled with soapy water (Dawn^®^ dish soap, Proctor & Gamble, Cincinnati, OH, USA and water) and placed in a triangle. Along with pan traps, yellow sticky card traps (Alpha Scents, Inc., Canby, OR, USA): yellow card-double sided, 20 × 15 cm (8 × 5.5 in.) were placed at every location. These traps were attached to garden stakes with clothes pins and were set adjusted to be even with plant height.

Sampling occurred between 8:00, when bowls and sticky cards were placed, and 19:30 when traps were collected. When traps were checked, the contents of each bowl were poured through a strainer and collected insects were placed in a labeled 50-mL plastic vial with 70% ethyl alcohol. Sticky card traps were placed in labeled plastic bags. For controls, four sampling locations were randomly chosen in a field that had only the resistant cultivar (Tx2783) and sampling protocols were the same as above.

After collection, the pan trap contents were stored in vials. The Hymenoptera from pan traps were sorted, washed, and classified to morphotype, and voucher specimens were pinned and labeled. Hymenoptera on sticky traps were viewed under a dissecting microscope and identified to morphospecies.

#### 2.2.2. Johnsongrass

The effects of SA presence on Hymenoptera were also tested in Johnsongrass. In total, there were seven trap locations with natural SA infestations and one site that did not have aphids that acted as a control. Four sites were monitored at the Cimarron Valley Research Station, Perkins, OK, USA, a small farm owned by Oklahoma State University (OSU) with a fifth site acting as a control. Two additional sites with Johnsongrass and SA infestations were monitored at the USDA-ARS research station and at the OSU Insect Adventure in Stillwater, OK, USA.

#### 2.2.3. Analyses

To examine the influence of aphids on Hymenoptera, chi-squared goodness of fit tests were used to compare all Hymenoptera collected in sorghum and Johnsongrass, respectively with significance being judged at *p* < 0.01. To estimate the effects of aphids on Hymenoptera diversity, morphospecies results from traps in plots with SA were compared with traps in control sites using Shannon-Wiener (diversity) and Simpson’s (evenness) indices [24]. Index values cannot be statistically compared, but higher values indicate higher overall diversity or evenness among morphospecies, respectively.

## 3. Results

### 3.1. Study 1

#### 3.1.1. Insects Observed Collecting or Consuming Pollen

Over the four-week period from September to October 2021, insects belonging to four genera were observed collecting or consuming pollen (Table 1; Figure 2) on the sweet sorghum population. The maize calligrapher (*Toxomerus politus*) was the most abundant on sorghum panicles, followed by honeybees (*Apis mellifera*), earwigs (*Doru taeniatum*), and bumblebees (*Bombus* sp.).

#### 3.1.2. Distribution of Insects and Morphological Traits

The distribution of insects observed per panicle ranged from 0–3 (mean = 0.37) for maize calligraphers, 0–2 (mean = 0.19) for honeybees, 0–6 (mean = 0.10) for earwigs, and 0–1 for bumblebees (mean = 0.009). Among the F_4_ lines, there were differences between number of total insects observed [F(24, 195) = 2.21, *p* = 0.0016]) on the sorghum panicles—but not for total bees or hoverflies. Line F_4_-20 had the greatest number of total insects (14 earwigs, 3 honeybees, and 1 hoverfly) whereas line F_4_-67 had the lowest number of total insects with 0 observed (Figure 3). It should be noted that line F_4_-20 also harbored most of the earwigs that were observed (14 out of 23 total), but the dataset contained too many zero observations to analyze statistically.

#### 3.1.3. Probability of Insect Observations by Date

Among the 50 lines, PI 257599 and 24 of the F_4_ lines had all three replicates flowering during the study period and thus could be sampled for insects. The probability of observing any insect on a sorghum panicle varied by date [F(3, 21) = 3.61, *p* = 0.030] with the probability being greater on the first date (14 September; *p* = 0.62) than on the last date (4 October; *p* = 0.11). However, other differences were not statistically significant. 

The probability of observing a bee (honeybees and bumblebees) on a sorghum panicle showed a borderline significance based on date [F(3, 21) = 3.01, *p* = 0.053] with a slightly greater probability (*p* = 0.068) of observing a bee on the third date (28 September; *p* = 0.28) than on the second date (21 September; *p* = 0.10). There was no date effect for the probability of observing a hoverfly on a sorghum panicle [F(3, 21) = 1.39, *p* = 0.273].

#### 3.1.4. Differences in F_4_ Morphological Traits

The F_4_ lines had a range for plant height and panicle length of 100–365 cm (mean = 249.9 cm) and 11–30 cm (mean = 22.5 cm), respectively (Appendix A). Plant height [t(16) = 418.62, *p* < 0.0001)] and panicle length [t(16) = 14.64, *p* = 0.0015] were significantly different among parental lines. PI 257599 was taller (average of 289.4 cm vs. 138.3 cm) and had a longer panicle length (average of 20.4 vs. 16.1 cm) than N109B. Differences in panicle length were observed among the F_4_ lines [F(24, 195) = 7.55, *p* < 0.0001]. Line F_4_-106 had the longest panicle whereas line F_4_-8 had the shortest panicle (Appendix A). Similarly, there were differences in plant height among the F_4_ lines [F(24, 195) = 47.78, *p* < 0.0001]. Line F_4_-46 was the tallest line whereas F_4_-20 was the shortest line (Appendix A). Despite plant differences, there were no significant effects of panicle length and plant height on the number of total insects, total bees, or hoverflies (not shown).

### 3.2. Study 2

Using both trapping methods, a total of 3940 Hymenoptera were collected with 2376 individuals from sorghum, and 1564 from Johnsongrass (Table 2 and Table 3). We identified similar families and morphospecies from sorghum and Johnsongrass when SA were present. A total of 29 identified families and 121 morphospecies were found in sorghum and 28 identified families and 116 morphospecies were collected from Johnsongrass.

The presence of SA strongly influenced the number of Hymenoptera collected in both sorghum and Johnsongrass. In sorghum, 1892 Hymenoptera were collected from plots with SA compared to 484 from uninfested plots (Table 2). The presence of aphids significantly increased the number of ants, halictid bees, and scelionid wasps. Aphids also significantly increased the presence of sphecid, diapriid, mymarid, encyrtid, and braconid wasps, but did not influence the presence of pompilids or mutillids (Table 2). Using pans and sticky traps, 1490 Hymenoptera were collected from Johnsongrass plots with SA and only 74 from uninfested plots. In Johnsongrass, aphids also increased the numbers of ants, halictid bees, braconid, sphecid, encyrtid, mymarid, and scelionid wasps but also increased the number of pompilid and figitid wasps (Table 3).

Pan traps collected more individual Hymenoptera than sticky traps (2588 compared to 1363, respectively) with about 2 times more individuals in pan traps placed in sorghum and Johnsongrass (Table 4). The highest number of morphospecies (100) were collected in pan traps in sorghum and the lowest number were collected from sticky traps placed in Johnsongrass (65). Both crop type and trap type influenced captures of Hymenoptera. Families that were most abundant in sorghum were Halictidae (*n* = 638), Scelionidae (*n* = 466) and Formicidae (*n* = 613). In Johnsongrass, Formicidae (*n* = 801) and Scelionidae (*n* = 312) were commonly collected. Pan traps collected more specimens of larger Hymenoptera (Apoidea, Pompiloidea, and Formicoidea) compared to sticky traps that collected smaller parasitic species (Scelionidae, Platygasteridae, and Mymaridae) (Table 4).

The Shannon-Wiener diversity indices were higher for Hymenoptera collected in sorghum with SA compared to uninfested sorghum for the majority of samples and the diversity increased through the duration of the experiment (Table 5). Simpson index values that are close to 1.0 represent equality in the number of captures among morphospecies. Among samples, Shannon-Wiener diversity for sorghum was moderate but overall evenness of diversity was poor, suggesting attraction of taxa to the honeydew resources (Table 5). In contrast to the results obtained from sorghum, the Shannon-Diversity indices of Johnsongrass were always greater for plots with SA. The Simpson index values also indicated dominance by a few taxa; however, the loss of control plots through natural infestation of SA limited our ability to interpret the data (Table 5).

## 4. Discussion

Both studies revealed a substantial number of Hymenoptera species utilizing resources from sorghum plants. Honeybees, bumblebees, and carpenter bees were all observed collecting sorghum pollen, while maize calligraphers, and earwigs were observed consuming sorghum pollen directly from the plant’s panicles (Figure 2; Table 1). Previously, honeybees have been documented collecting sorghum pollen [12]. Sorghum pollen has between 13–26% raw protein and is a source of essential amino acids, fat (4.75%), carbohydrates (53%), and fiber (13%) [25,26]. Siede et al. [11] found that a diet of sorghum pollen alone was sufficient for bee brood. To our knowledge, this is the first documentation of bumblebees and carpenter bees collecting sorghum pollen.

For the insects consuming pollen directly from the panicle, only the maize calligrapher has been previously documented feeding on sorghum pollen flowers [27]. The maize calligrapher can utilize grass pollen of sorghum and maize (*Zea mays*) over its entire lifecycle [27,28]. However, earwigs (*Doru taeniatum*) have been previously documented feeding on maize pollen [29], but not sorghum.

The insects that consume pollen directly from the panicle are unlikely to impact sorghum cross-pollination. Despite their abundance, maize calligraphers have been shown to carry very little pollen [30]. The presence of earwigs may benefit the sorghum plant because *Doru taeniatum* is a predator of eggs and small larvae of the fall armyworm [31].

There was a slight date effect for observing bees collecting pollen from sorghum panicles, likely explained by weather differences. On 21 September 2021 the sky was overcast and rain was expected later in the afternoon, whereas the other three dates had clear skies. For honeybees, foraging is often reduced if solar radiation falls below 300 W/m^2^ and moderate wind is present [32]. Additionally, there were differences in total insects among the F_4_ lines. Line F_4_-20 had an abundance of earwigs present in the sorghum panicles and no insects were observed for any of the replicates of line F_4_-67 plots. The plant line with no observable insects may be unattractive due to odor or texture or may be a result of late flowering. On 4 October 2021, only two lines were rated and these two lines were among the five lowest lines for total insects observed per panicle. It is possible with cooling nighttime temperatures that pollen shed was impacted and may have also contributed to a reduced number of insects consuming or collecting pollen. Brooking [33] determined that nighttime temperatures of 10–14 °C during microsporogenesis impacted sorghum pollen viability but it is unknown how cooling nighttime temperatures impact the amount of pollen shed. Alternatively, shortening daylengths or drought (no rain from 23 September–4 October) may have impacted insect visitation (Appendix A).

There was a lack of evidence that panicle length and plant height of sorghum influenced the total number of observed insects. In previous research, flower visitation increased with inflorescence size exclusively for small hoverflies, and flower visitation increased with height for individual hoverfly species in flower rich grassland patches made up of primarily *Centaurea scabiosa*, *Galium mollugo*, *Galium verum*, *Agrimonia eupatoria*, and *Daucus carota* [34]. We had hypothesized sorghum with longer panicles which contained more anthers, or sorghum that was shorter and less likely to sway in the wind, may be more attractive to insects consuming or collecting pollen. However, our data did not support this.

Additionally, the effect of systemic insecticides such as Sivanto^®^ (17.09% flupyradifurone) and Transform ^®^ (50% sulfoxaflor; Corteva Agriscience, Indianapolis, IN, USA) on sorghum plants to control aphids should be further examined after these findings. Evidence from previous studies and our current study reveal that honeybees, bumblebees, and carpenter bees are collecting pollen from sorghum, and pollinator health should be considered even if the host is primarily wind-pollinated. Sivanto^®^ and Transform^®^ have been found to be toxic to honeybees [35,36] and bumblebees [37,38,39]. However, Sivanto^®^ is less toxic compared to Transform for honeybees [36]. Chakrabarti et al. [36] applied a dosage directly to honeybees at 400 mL/0.4 Ha (14 oz/acre) of Sivanto and 6 mL/0.4 Ha (2.25 oz/acre) of Transform. The manufacturers recommend using Sivanto at 118–414 mL/Ha (4–14 oz/acre) and using Transform at 22–44 mL/Ha (0.75–1.5 oz/acre) to control SA on sorghum [40,41]. These observations also emphasize the importance of applying pesticides at times when plants, including grasses, are not flowering to reduce risk to pollinators.

Further, the sticky trap survey revealed that flying, solitary Hymenoptera provide a strong indication of the influence of SA on diversity. These Hymenoptera can be attracted to the sugar resource [15], the prey [6], or to the host plants [17]. Because all flying Hymenoptera require sugars to fuel flight, it could be assumed that they are taking advantage of the resource; however, it is unclear whether they favor, or even benefit from, consuming honeydew. Parasitoids have been known to increase in diversity and abundance as prey organisms increase in prevalence [42]. These species may respond to SA for both nutrients or breeding resources. The Apoidea, which are nectar and pollen feeders, provide the clearest example of a positive influence of SA honeydew on Hymenoptera. The collection of 12 times more Halictidae in plots with SA in sorghum and 26 times more Halictidae in Johnsongrass with SA supports the conclusion that the honeydew produced by SA is being used by Halictidae. Almost all ants (*n* = 1414) were collected from pan traps compared to sticky traps (*n* = 16). Because ants are all eusocial insects that forage collectively, their influence on diversity and total number of Hymenoptera collected should also be viewed with caution. Ant diversity in sorghum appears limited compared with those found in Johnsongrass. The Johnsongrass plots were sampled with traps placed on the field edges away from the managed agroecosystems. The difference in the number of morphospecies collected in pan traps supports this observation (*n* = 3 in sorghum, *n* = 11 in Johnsongrass). Additionally, 6 morphotypes were collected on sticky traps in Johnsongrass while none were collected on sorghum sticky traps.

The trapping methods used in this study did not capture all of the Hymenoptera that we observed visiting aphid honeydew, especially for larger species. We captured relatively few European honeybees (*Apis mellifera*) despite regularly observing large numbers visiting honeydew. Additionally, species including *Pepsis* spp., tarantula hawk wasp, and the cicada killer, *Sphecius speciosus*, were observed in the fields, but never captured in pan traps or on yellow sticky traps. Future research should focus on creating transects within sorghum and Johnsongrass plots and using visual counts (as in Study 1) or netting larger Hymenoptera species. Active netting of individuals would ensure that the larger Hymenoptera not normally captured with pan traps or sticky traps could be evaluated [43].

Sorghum, like most grasses, does not produce nectar, and thus, commercial sorghum fields provide few apparent sugar resources for Hymenoptera. Areas with Johnsongrass that form dense monocultures likewise provide few apparent resources. In contrast to expectations, a diversity of Hymenoptera were present in areas with Johnsongrass, though strongly influenced by a few common species (Table 4). When SA was present, the number of individual Hymenoptera increased and there was a trend of increasing diversity for the duration of this 6-week study. Between late July and September, floral resources may become scarcer with high temperatures (>40 °C) and lack of rainfall. Thus, SA honeydew could play a crucial role in maintaining some Hymenoptera species that act as pollinators, predators, or parasitoids.

## 5. Conclusions

The majority of contemporary research supports the hypothesis that as plant communities diversify, the fauna also diversifies [5,44]. Not only does diversity improve with a diverse plant community, but crop health and yield also benefits from plant diversity [45]. Crop diversity benefits natural enemies that may provide pest control in agroecosystems [5]. In addition, diverse cover crops improve plant biomass and nutrient availability within the soil even after the cover crops are harvested from the system [46]. Despite these documented benefits, large-scale monocultures are widely planted because of efficiencies gained in cultivation and harvest. Thus, finding novel ways to provide ecosystem benefits that promote native organism populations are critical to sustainable production.

Calderone [47] estimated the economic value of pollinators for U.S. agricultural systems to be $15.12 billion (2009) and $12 billion (2004) for direct and indirectly pollinated crops, respectively. Much of the value of pollination is attributed to the management of the European honeybee. Honeybees are documented to collect pollen from sorghum, but this is the first experimental evidence of honeybees also collecting honeydew from an invasive pest that form large colonies and produce large quantities of waste. Our studies show that susceptible sorghum could provide a food source to at least 29 families of Hymenoptera and other beneficial insects and could be promoted to homeowners as a valuable landscape planting for preserving these insects. A limitation of these studies is that they were conducted for one year and in one location for each study. Future research should examine if SA could be managed to provide resources to benefit both managed beehives and native biodiversity by planting a strip of susceptible sorghum at the field edge and planting the rest of the field with a SA-resistant cultivar. The commercial use of pesticides on wind-pollinated grasses should also be further researched. Because of the utilization of sorghum plants by multiple bee species, current pesticide practices may be harming beneficial pollinators.

## Figures and Tables

**Figure 1 insects-13-01152-f001:**
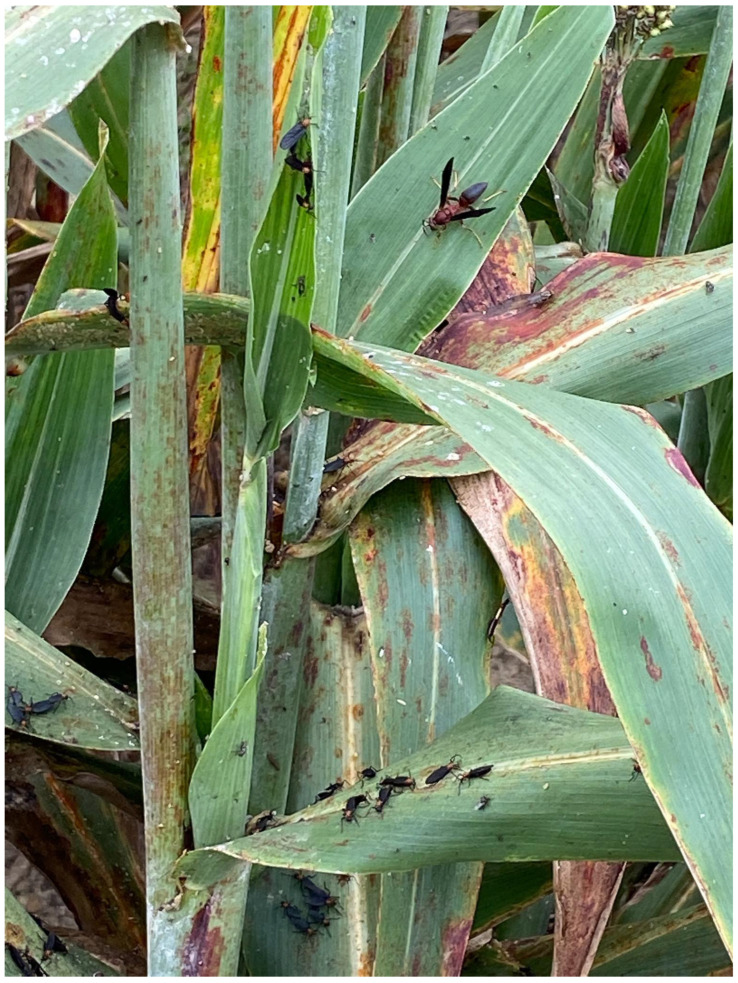
Hymenoptera and Diptera attracted to honeydew on grain sorghum infested with sorghum aphid, *Melanaphis sorghi*.

**Figure 2 insects-13-01152-f002:**
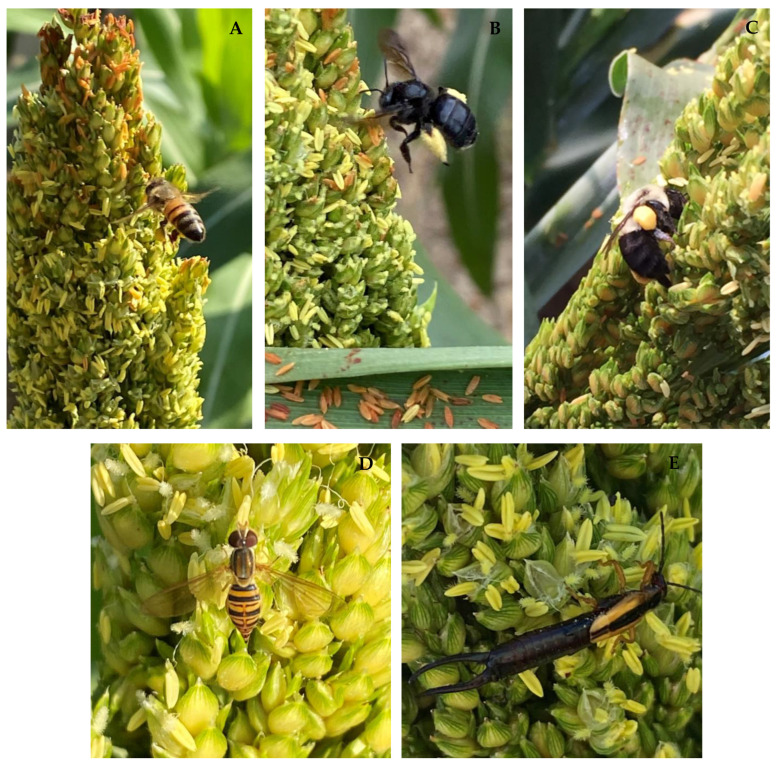
Top panel, (**A**) honeybee (*Apis mellifera*), (**B**) carpenter bee (*Xylocopa micans*), and (**C**) bumblebee (*Bombus* sp.) collecting sorghum (*Sorghum bicolor*) pollen. Bottom panel, (**D**) a hoverfly (*Toxomerus politus*) and (**E**) earwig (*Doru taeniatum*) consuming sorghum pollen.

**Figure 3 insects-13-01152-f003:**
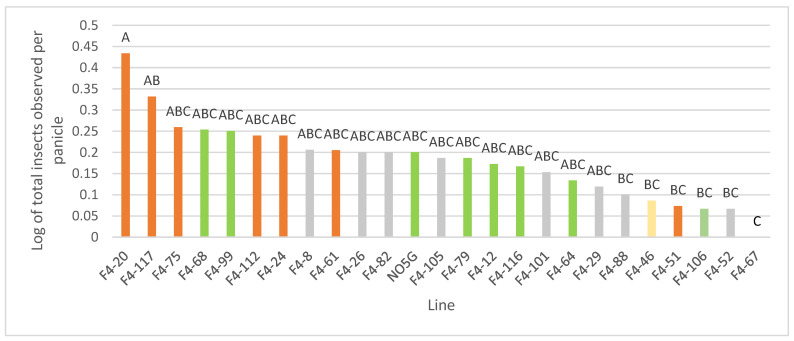
Insects (log10) observed per panicle for the N109A × PI 257599 F_4_ sorghum lines in Tifton, GA 2021. NO5G is PI 257599 (the common name is No. 5 Gambela). Means with the same letter are not different at α = 0.05. Orange, grey, green, and gold shaded lines were phenotyped on 14 September 2021, 21 September 2021, 28 September 2021, and 4 October 2021, respectively. Line F_4_-67 was phenotyped on 4 October 2021.

**Table 1 insects-13-01152-t001:** Insects observed collecting or consuming pollen on a sweet sorghum population over four rating dates in September–October 2021.

Family	Organism	Common Name	Total No. Observed
Syrphidae	*Toxomerus politus*	Maize calligrapher	83
Apidae	*Apis mellifera*	Honeybee	43
Forficulidae	*Doru taeniatum*	Earwig	23
Apidae	*Bombus* sp.	Bumblebee	2
Apidae ^y^	*Xylocopa micans*	Carpenter bee	1

^y^ Observed on sorghum border plots only.

**Table 2 insects-13-01152-t002:** Comparison of Hymenoptera families and morphospecies collected in sorghum with sorghum aphids and un-infested checks using pan traps and sticky cards.

		Sorghum			
		With Aphids		Without Aphids	
Superfamily	Family	# of Morphospecies	# of Individuals	# of Morphospecies	# of Individuals
Apoidea					
	** Halictidae	12	618	6	48
	Apidae	9	10	1	1
	Andrenidae	1	5	1	1
	* Sphecidae	19	41	4	6
	Megachilidae	0	0	0	0
Pompiloidea					
	Pompilidae	8	34	5	37
	Mutillidae	1	4	0	0
Chrysidoidea					
	Chrysididae	1	1	0	0
	Bethylidae	1	5	2	8
	Dryinidae	1	1	0	0
Tiphioidea					
	Tiphiidae	1	7	1	3
	Sierolomorphidae	1	1	0	0
Ichneumonoidea					
	Ichnuemonidae	2	2	1	1
	* Brachonidae	7	17	2	3
Chalcidoidea					
	Aphelinidae	2	5	1	1
	Chalcidae	3	7	0	0
	** Encyrtidae	6	89	5	26
	Eupelmidae	1	1	1	1
	Eurytomidae	2	3	0	0
	** Mymaridae	4	48	4	9
	Pteromalidae	7	26	1	50
	Torymidae	1	6	1	3
	Eulophidae	2	3	0	0
	Perilampidae	1	1	1	1
Cynipoidea					
	Figitidae	4	17	1	5
Diaprioidea					
	* Diapriidae	2	15	1	1
Ceraphronoidea					
	Ceraphronidae	1	2	1	1
Platygastroidea					
	Platygastridae	3	13	2	11
	** Scelionidae	12	376	8	179
Formicoidea					
	** Formicidae	3	528	3	85
Evanioidea					
	Evaniidae	0	0	0	0
Unknown		3	6	3	3
TOTALS		121	1892	56	484

* Chi-squared goodness of fit *p* < 0.01. ** Chi-squared goodness of fit *p* < 0.001.

**Table 3 insects-13-01152-t003:** Comparison of Hymenoptera families and morphospecies collected in Johnsongrass infested with sorghum aphids and un-infested checks using pan and sticky traps.

		Johnsongrass			
		With Aphids		Without Aphids	
Superfamily	Family	# of Morphospecies	# of Individuals	# of Morphospecies	# of Individuals
Apoidea					
	** Halictidae	13	78	3	3
	Apidae	2	2	0	0
	Andrenidae	3	3	0	0
	* Sphecidae	6	10	0	0
	Megachilidae	1	1	0	0
Pompiloidea					
	* Pompilidae	9	16	2	1
	Mutillidae	2	16	0	0
Chrysidoidea					
	Chrysididae	1	1	0	0
	Bethylidae	4	9	1	7
	Dryinidae	0	0	0	0
Tiphioidea					
	Tiphiidae	1	4	0	0
	Sierolomorphidae	0	0	1	1
Ichneumonoidea					
	Ichnuemonidae	6	9	0	0
	* Brachonidae	6	12	2	2
Chalcidoidea					
	Aphelinidae	1	1	0	0
	Chalcidae	3	4	0	0
	** Encyrtidae	6	31	0	0
	Eupelmidae	2	2	0	0
	Eurytomidae	1	3	0	0
	** Mymaridae	6	36	3	4
	** Pteromalidae	3	7	2	2
	Torymidae	1	2	0	0
	Eulophidae	0	0	0	0
	Perilampidae	0	0	0	0
Cynipoidea					
	** Figitidae	3	22	1	1
Diaprioidea					
	Diapriidae	3	6	1	1
Ceraphronoidea					
	Ceraphronidae	2	33	0	0
Platygastroidea					
	Platygastridae	2	19	0	0
	** Scelionidae	13	365	3	23
Formicoidea					
	** Formicidae	11	789	5	28
Evanioidea					
	Evaniidae	1	1	0	0
Unknown		4	8	1	1
TOTALS		116	1490	25	74

* Chi-squared goodness of fit *p* < 0.01. ** Chi-squared goodness of fit *p* < 0.001.

**Table 4 insects-13-01152-t004:** Comparison of trapping method in sorghum and Johnsongrass.

		Sorghum				Johnsongrass			
		Pan		Sticky		Pan		Sticky	
Superfamily	Family	Types	#	Types	#	Types	#	Types	#
Apoidea									
	Halictidae	12	638	5	28	9	68	6	13
	Apidae	9	11	0	0	2	2	0	0
	Andrenidae	1	6	0	0	3	3	0	0
	Sphecidae	19	42	3	5	5	10	0	0
	Megachilidae	0	0	0	0	1	1	0	0
Pompiloidea									
	Pompilidae	9	64	1	7	12	18	0	0
	Mutillidae	1	2	1	2	4	12	1	4
Chrysidoidea									
	Chrysididae	1	1	0	0	1	1	0	0
	Bethylidae	1	3	2	10	3	5	2	11
	Dryinidae	1	1	0	0	0	0	0	0
Tiphioidea									
	Tiphiidae	1	7	1	3	1	1	1	3
	Sierolomorphidae	1	1	0	0	0	0	1	1
Ichneumonoidea									
	Ichnuemonidae	1	1	2	2	5	7	2	2
	Brachonidae	4	7	5	19	3	3	3	11
Chalcidoidea									
	Aphelinidae	1	2	1	4	1	1	0	0
	Chalcidae	2	6	1	1	0	0	3	4
	Encyrtidae	4	9	6	107	1	4	6	28
	Eupelmidae	1	1	1	1	2	3	0	0
	Eurytomidae	1	1	2	2	0	0	1	3
	Mymaridae	1	2	4	55	0	0	6	40
	Pteromalidae	7	8	6	68	2	2	4	7
	Torymidae	0	0	1	10	0	0	1	2
	Eulophidae	2	2	2	1	0	0	0	0
	Perilampidae	0	0	1	2	0	0	0	0
Cynipoidea									
	Figitidae	3	11	3	11	3	7	2	16
Diaprioidea									
	Diapriidae	2	12	2	4	3	5	2	2
Ceraphronoidea									
	Ceraphronidae	1	1	1	2	2	8	1	25
Platygastroidea									
	Platygastridae	0	0	3	24	1	1	2	18
	Scelionidae	10	89	11	466	12	76	11	312
Formicoidea									
	Formicidae	3	613	0	0	11	801	6	16
Evanioidea									
	Evaniidae	0	0	0	0	1	1	0	0
Unknown									5
	Unidentified	1	3	3	6	1	4	3	
Total Families		26		24		24		20	
TOTAL		100	1544	68	840	89	1044	65	523

**Table 5 insects-13-01152-t005:** Diversity indices of Hymenoptera collected using pan or sticky traps from sorghum and Johnsongrass infested with sorghum aphid or uninfested (control).

Shannon-Weiner Diversity Index (Sorghum)				
	Pan		Sticky	
	With Aphids	Control	With Aphids	Control
31 July 2019	1.914	0.910	2.403	1.991
17 August 2019	2.037	2.239	2.123	1.120
2 September 2019	2.147	1.526	2.100	0.674
9 September 2019	2.179	1.826	2.029	1.750
16 September 2019	2.252	1.229	2.513	2.594
27 September 2019	2.617	2.579	2.587	2.332
**Simpson Diversity Index (Sorghum)**				
	**Pan**		**Sticky**	
	**With Aphids**	**Control**	**With Aphids**	**Control**
31 July 2019	0.749	1.000	0.884	0.810
17 August 2019	0.780	0.871	0.732	0.612
2 September 2019	0.794	0.751	0.836	0.475
9 September 2019	0.767	0.772	0.739	0.810
16 September 2019	0.841	0.638	0.875	0.931
27 September 2019	0.840	0.953	0.868	0.890
**Shannon-Weiner Diversity Index (Johnsongrass)**				
	**Pan**		**Sticky**	
	**With Aphids**	**Control**	**With Aphids**	**Control**
2 August 2019	2.509	1.300	1.888	1.851
26 August 2019	2.259	1.600	2.082	1.709
6 September 2019	2.177	N/A	2.376	N/A
15 September 2019	2.039	N/A	2.695	N/A
23 September 2019	2.387	N/A	2.587	N/A
**Simpson Diversity Index (Johnsongrass)**				
	**Pan**		**Sticky**	
	**With Aphids**	**Control**	**With Aphids**	**Control**
2 August 2019	0.843	0.891	0.633	0.842
26 August 2019	0.827	0.875	0.749	0.838
6 September 2019	0.808	N/A	0.897	N/A
15 September 2019	0.781	N/A	0.903	N/A
23 September 2019	0.819	N/A	0.883	N/A

## Data Availability

The data are available from the corresponding author upon request.

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
