# Peer review of "Insect Feeding on Sorghum bicolor Pollen and Hymenoptera Attraction to Aphid-Produced Honeydew"

_insects, 2022, doi:10.3390/insects13121152_

Round 1

Reviewer 1 Report

Dear authors,

generally I consider your manuscript well-written and interesting, with only minor issues to improve.

These include:

L93: please describe here what "Sivanto" is (it is described later in the discussion, but should be done here already)

Plant material: Is there any specific reason why the 50 F4-lines were chosen for this study? Maybe because they segregate for plant height and panicle size? This should be stated.

Can you provide weather data? Temperature records could explain the decline of insect visits on sorghum at later stages (see below).

Discussion:

It is interesting that you observe different attractivity of sorghum lines to insects. You have found insect visits being higher at the beginning of the experiment, and assume this was due to weather. Probably the weather conditions had impact on the pollen shedding of the sorghum, which is likely to decrease in fall with colder temperatures. From my point of view, the amont of pollen released by the different sorghum lines is likely to be the decisive factor for their number of insect visits, i. e. attractiveness, rather than plant height or panicle size. Maybe the line with no insect visits was just an extremely poor pollen shedder, with cold temperature-induced male sterility. Was pollen shedding scored? These aspects should be complemented in the discussion. 

L289-292: I would rather say insects do not help sorghum self pollination (under good weather conditions, i. e. good pollen amount and fertility, self fertilty of sorghum is no problem.), but can help at cross pollination, i. e. transferring pollen from one panicle to another, especially when plants are stressed and pollen is scarce. This can even have positive yield effects and has been shown by Siede et al. 2021.

One limitation of the manuscript is that for both studies you have only one environment (location x year combination). This should be critically acknowledged.

Author Response

Thank you for your thorough review and excellent suggestions.  We revised according to your suggestions and added supplementary data on weather.

Here is our response:

L93: please describe here what "Sivanto" is (it is described later in the discussion, but should be done here already)

Response: Correction made

Plant material: Is there any specific reason why the 50 F4-lines were chosen for this study? Maybe because they segregate for plant height and panicle size? This should be stated.

Response: Correction made

Can you provide weather data? Temperature records could explain the decline of insect visits on sorghum at later stages (see below).

Results: Weather data was added and this is an important point. Temperatures were quite warm during this study (the lowest was on Sept 24, 2021 at 54.7 °F). It was cooler in mid September than early October.  There was no rain from Sept 23-Oct 4, 2021.

Discussion:

It is interesting that you observe different attractivity of sorghum lines to insects. You have found insect visits being higher at the beginning of the experiment, and assume this was due to weather. Probably the weather conditions had impact on the pollen shedding of the sorghum, which is likely to decrease in fall with colder temperatures. From my point of view, the amont of pollen released by the different sorghum lines is likely to be the decisive factor for their number of insect visits, i. e. attractiveness, rather than plant height or panicle size. Maybe the line with no insect visits was just an extremely poor pollen shedder, with cold temperature-induced male sterility. Was pollen shedding scored? These aspects should be complemented in the discussion. 

Response: Pollen shedding was not scored but is an important aspect that deserves further exploration. We pulled the weather data from the location for Sept 14, 2021 to Oct 4, 2021 and it was quite warm the first week of Oct. (lows were in the 60s and highs were in the 80s F). I looked up the literature on what is known about pollen shedding and temperature for sorghum. During microsporogenesis if the nighttime temperatures are below14 C pollen viability is reduced (Brooking 1979). In control plants it takes 12-15d from microsporogenesis to the flag leaf emergence and then many more days to pollen shedding. 14C is 57.2 F and Sept 23-24/26, 2021 had nighttime temperatures that fell below 57.2 F. Yet, it should impact the sorghum that flowers long after this date, so Oct 4, 2021 was only 11 d after these critical temperatures. Although cooler temperatures impact sorghum viability I’m not sure how it impacts how much pollen is actually shed. The temperatures in Georgia are very warm and we added supplementary figure 1 to help readers interpret the data.

L289-292: I would rather say insects do not help sorghum self pollination (under good weather conditions, i. e. good pollen amount and fertility, self fertilty of sorghum is no problem.), but can help at cross pollination, i. e. transferring pollen from one panicle to another, especially when plants are stressed and pollen is scarce. This can even have positive yield effects and has been shown by Siede et al. 2021.

Response: The sentence on self pollination was deleted.

One limitation of the manuscript is that for both studies you have only one environment (location x year combination). This should be critically acknowledged.

Response: Yes, we added this statement to the Conclusion as a limitation to our studies. (lines 395-396)

Reviewer 2 Report

This paper demonstrates that susceptible sorghum may be acting as a food source for families of Hymenoptera/other insect species. More insects were found on aphid infested sorghum compared to non-infested sorghum, which was also found in infested versus non-infested Johnsongrass. I believe the results of the study support the conclusions, and have potential applications for land management strategies which could promote insect biodiversity in agricultural settings. I suggest only a few minor changes to the manuscript, and congratulate the authors on their research.

Abstract

Good abstract providing a nice overview of the work, with very few improvements to make.

Line 23- 'suggest' rather than 'show' might be better? You've put in the conclusions 'sorghum could provide a food source' which I think is a better way of putting it.

Line 24- 'SA provides energy from honeydew', change to 'could provide energy from honeydew'.

Introduction

Good background of literature and introduction to current study. Aims are clearly presented in the final paragraph, and seems like a thorough literature review has been performed. Minor suggestions below, which are typographical.

Line 45- comma instead of hyphen 

Line 54- comma after 'grasses'

Line 74- replace semi-colon with 'and'

Line 84- Italicise 'Melanaphis sorghi'

Methods

Robust and clear overview of methods, especially the statistical analysis section. I like the structure of the paper and the clear descriptions of 'study 1' and 'study 2'.

Line 85- colon after 'Study 1', to keep consistent

Line 93- I would change 'sprayed with Silvanto' to 'sprayed with Silvanto, an insecticide' or something similar

Line 168- full stop after 'OK'

Results

Clear results. Main amendment is rewording of the caption beneath Figure 1, and a few other minor points.

Line 184- What is the significance of the carpenter bee visit? Not sure it needs a whole sentence unless theres something significant about the visit which gets discussed later, which i'm not sure it does

Line 200- Might need to reword 'Number of total insects observed', as this sounds like the absolute number of insects visiting per panicle. The data have been log transformed, so the y axis isn't really showing the absolute number of insects visiting. You could either back transform the data and show the absolute number of insects visiting, or update the figure legend to say it's the log10 of total insects observed per panicle. 

Discussion

Results clearly placed into the context of the background literature. A few examples of where the authors have used subjective terms i.e. 'far less toxic', 'moderately high', which are a bit vague and could be reworded. Similarly, mention of 'high temperatures', it would be good to clarify what these temperatures were (approximately). 

Line 281- brackets after [23]?

Line 230- Reword 'far less toxic' as its quite subjective- significantly less toxic, if the appropriate statistics were performed in their study? Or just 'less toxic' if not

Line 328/329- Include reference(s) about Hymenoptera being attracted to sugar resource, prey, host plant?

Line 360- 'moderately high' compared to what? Quite subjective 

Line 364- What were the high temperatures?

Conclusions

In general, a good summary of the findings, with a particular focus on how these findings could be used to inform land management strategies to preserve insect biodiversity. However, I think this section is quite long. I think lines 368-383 could actually have a place either in the introduction or discussion. The second paragraph could start with 'Our studies show that susceptible sorghum....' and still be a nice concluding paragraph.

Author Response

Thank you for your thorough review.  We made all changes, except for shortening the conclusions. 

Reviewer 3 Report

The manuscript is a good work. Authors clearly indicate the novelty of their manuscript. Some minor revision are needed to meet the journal's standards

Line 47

When you first mention an insect species, you must refer the order and the family, e.g. Apis mellifera (Hymenoptera, Apidae)

Lines 48-49

Since you refer to multiple species, you have to use the plural spp. instead of sp.

Line 50

Use the MDPI citation format

Line 55 Honeydew is often characterized as damage [...]

You must reformulate this sentence. Maybe you could use considered instead

Line 108

Times should be written using the 24-hour clock with a colon between the hours and minutes (MDPI Style guide 6.3). Make it 8:00 and 10:30 (without AM)

Lines 109-110

Dates should be written with the format day (as a digit) month (as a word) year (four digits), e.g., 14 September 2021 (MDPI Style guide 6.3).

Line 116

As in Lines 109-110

Line 151

As in Line 108

Lines 182-185

As in Line 47

Line 185

As in Lines 109-110

Line 203

As in Lines 109-110

Line 212

As in Lines 109-110

Table 5

You have to reform all the dates 7/31/2019 to 31/7/2019 etc

Line 281

Delete the parentesis. et al. should be in italics

Line 286

Zea mays  in italics. Also, why do you use the authority name (L.) only in here and not in the rest of the manuscript? You have to make it all either with the authority name or without.

Line 296

As in Lines 109-110

Line 303

As in Lines 109-110

Line 310

scabiosa with lowercase s

Lines 392-400

Funding must be in a different paragraph. Also, you must declare if there is any conflict of interest, and the authors contibutions. See the instructions for authors.

References

You must reform the whole reference list to meet the MDPI standards. Journal names, scientific names and latin phrases (e.g. et al.) must be in italics. Also some tiltes are in capitals(e.g lines 409, 418). Scientific names must be written  with a lowercase in the species name  (e.g. lines 423, 457, 463)

Author Response

Thank you for all of the editing to make our paper consistent with MDPI guidelines.  We made all of your suggested changes.
